# Blueprinting extendable nanomaterials with standardized protein blocks

Timothy F. Huddy[1,2,11], Yang Hsia[1,2,11], Ryan D. Kibler[1,2,11], Jinwei Xu[1,2,11], Neville Bethel[1,2], Deepesh Nagarajan[3], Rachel Redler[4], Philip J. Y. Leung[1,2,5], Connor Weidle[1,2], Alexis Courbet[1,2,6], Erin C. Yang[1,2,7], Asim K. Bera[1,2], Nicolas Coudray[4,8,9], S. John Calise[1], Fatima A. Davila-Hernandez[1,2], Hannah L. Han[1,2], Kenneth D. Carr[1,2], Zhe Li[1,2], Ryan McHugh[1,2], Gabriella Reggiano[1,2], Alex Kang[1,2], Banumathi Sankaran[10], Miles S. Dickinson[1], Brian Coventry[1,2], T. J. Brunette[1,2], Yulai Liu[1,2], Justas Dauparas[1,2], Andrew J. Borst[1,2], Damian Ekiert[4,8], Justin M. Kollman[1], Gira Bhabha[8] & David Baker[1,2,6 ✉]

A wooden house frame consists of many different lumber pieces, but because of the regularity of these building blocks, the structure can be designed using straightforward geometrical principles. The design of multicomponent protein assemblies, in comparison, has been much more complex, largely owing to the irregular shapes of protein structures[1]. Here we describe extendable linear, curved and angled protein building blocks, as well as inter-block interactions, that conform to specified geometric standards; assemblies designed using these blocks inherit their extendability and regular interaction surfaces, enabling them to be expanded or contracted by varying the number of modules, and reinforced with secondary struts. Using X-ray crystallography and electron microscopy, we validate nanomaterial designs ranging from simple polygonal and circular oligomers that can be concentrically nested, up to large polyhedral nanocages and unbounded straight 'train track' assemblies with reconfigurable sizes and geometries that can be readily blueprinted. Because of the complexity of protein structures and sequence–structure relationships, it has not previously been possible to build up large protein assemblies by deliberate placement of protein backbones onto a blank three-dimensional canvas; the simplicity and geometric regularity of our design platform now enables construction of protein nanomaterials according to 'back of an envelope' architectural blueprints.

There has been considerable recent progress in the design of protein nanomaterials including cyclic oligomers[2–4], polyhedral nanocages[5–8], one-dimensional (1D) fibres[9,10], 2D sheets[11,12] and 3D crystals[10,13] by docking together[8] or fusing[6,14] protein monomers or cyclic oligomers. Although powerful, these methods have two limitations that arise from the irregularity of almost all protein structures. First, because the shapes of the constituent components are generally complex, they cannot be assembled into higher-order structures on the basis of simple geometric principles; instead, large-scale sampling calculations are required to identify shape-complementary interactions for each case, and there is no guarantee that designable interfaces can be found. Second, as for the myriad protein complexes in nature, the size of a designed protein assembly cannot be readily scaled; it is nearly impossible to make a smaller or larger but otherwise nearly identical version of assemblies generated using current design methods. By contrast, designed materials that extend along just one dimension, such

as α-helical coiled coils and repeat proteins, can be grown or shrunk by simply varying the length of the chain. There is a rich history of designing coiled coils using simple geometric principles; this extensibility and designability have made them widely used constituents of designed protein materials[15].

We set out to develop a general approach for designing expandable higher-order protein nanomaterials with the simplicity and programmability of coiled-coil engineering. We reasoned that if a modular and regular toolkit of building blocks and interactions could be generated consisting of linear building blocks constructed from (1) repeating sequence elements that extend without twisting as additional sequence repeats are added (Fig. 1b,c(top),d), (2) curved building blocks that trace out arcs of circles of different radii (Fig. 1c(bottom),d) and (3) non-covalent arrangements that hold two building blocks in pre-specified relative orientations (Fig. 1e), then building up new nanostructures could in principle be carried out by inspection in a manner

[1]Department of Biochemistry, University of Washington, Seattle, WA, USA. [2]Institute for Protein Design, University of Washington, Seattle, WA, USA. [3]M.S. Ramaiah University of Applied Sciences, Bengaluru, India. [4]Department of Cell Biology, NYU School of Medicine, New York, NY, USA. [5]Molecular Engineering and Sciences Institute, University of Washington, Seattle, WA, USA. [6]Howard Hughes Medical Institute, University of Washington, Seattle, WA, USA. [7]Biological Physics, Structure and Design, University of Washington, Seattle, WA, USA. [8]Applied Bioinformatics Laboratories, NYU School of Medicine, New York, NY, USA. [9]Division of Precision Medicine, Department of Medicine, NYU Grossman School of Medicine, New York, NY, USA. [10]Molecular Biophysics and Integrated Bioimaging, Berkeley Center for Structural Biology, Lawrence Berkeley National Laboratory, Berkeley, CA, USA. [11]These authors contributed equally: Timothy F. Huddy, Yang Hsia, Ryan D. Kibler, Jinwei Xu. ✉e-mail: dabaker@uw.edu

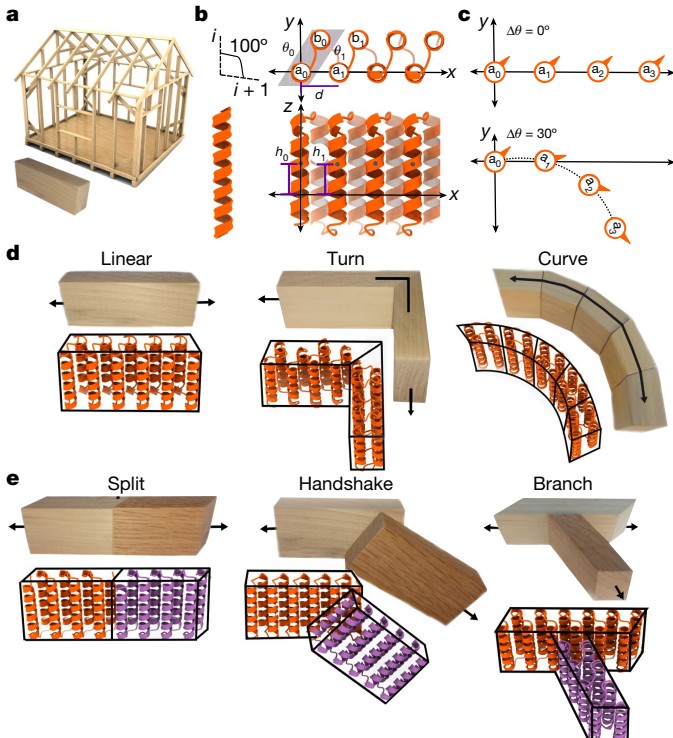

**Fig. 1 | Overview of THR protein blocks and interaction modules. a**, Building a house frame from standardized wooden building blocks. **b**, THR internal geometry. Blocks are constructed from idealized straight α-helices with an angle of rotation between adjacent helices of $\Delta\theta$; the remaining degrees of freedom that contribute to the repeat trajectory are also indicated. **c**, Changing $\Delta\theta$ (while holding the other parameters constant) specifically changes the curvature of the repeat trajectory. **d**, Single-chain THR modules. **e**, THR interaction modules. Image of house frame (**a**, top) by mmaxer, Can Stock Photo.

analogous to blueprinting a house frame (Fig. 1a). Furthermore, as with a house frame, the regular structures of the constituent building blocks could enable scaling the dimensions of the final architecture (area or volume) by simply altering the size of the constituent monomers, and structural reinforcement by placement of additional buttressing elements.

## Design of twistless helix repeats

Natural and previously designed proteins exhibit a wide range of helical geometries with local irregularities, kinks and deviations from linearity[16] that make it difficult to achieve the properties illustrated in Fig. 1 that enable simple nanomaterial scaling (beyond the one dimension accessed by varying the number of repeats in a repeat protein or coiled coil). To achieve these properties, we designed a series of new building blocks constructed from ideal α-helices with all helical axes aligned. Restricting helical geometry to ideal straight helices with zero helical twist in principle considerably limits what types of structure could be built, but this is more than compensated by the great simplification of downstream material design, as illustrated below. We construct twistless helix repeat (THR) protein blocks from identical straight α-helices (typically 2–4 helices in each unit); the length of the blocks can be varied simply by varying the number of repeat units. In contrast to existing natural and designed repeat proteins[17], THRs are constructed to enable modular nanomaterial design: linear blocks are perfectly straight, allowing nanomaterials to be extended and contracted with no alteration in the angles between the constituent monomers; curve blocks have smoothly curving trajectories that stay in-plane; and turn as well as interaction modules enable placement of

two blocks in precise relative orientations with angles appropriate for regular material design.

We blueprint THRs by explicit placement of these straight helix structural elements using an extension of the principles used in coiled-coil and helical bundle design[16,18]. A first helix $a_0$, part of the zeroth repeat, is placed at the origin and aligned to the $z$ axis. A copy of $a_0$ called $a_1$ is then placed at a new location to set the rigid body transformation between the zeroth and first (and all subsequent) repeat units. After this, any other helices ($b_0$, $c_0$ …) that will be part of the repeating unit are placed as appropriate between $a_0$ and $a_1$ to provide more helices to pack against for stability, and the helices are connected with loops[19]; repetition of this basic unit then generates backbones with the desired geometries[17] (Fig. 1b,c). As the helices are perfectly straight and parallel to the $z$ axis, the overall repeat protein trajectory is fully defined by the following transformation parameters from $a_0$ to $a_1$: the distance of displacement in the $x$–$y$ plane from helical axis to helical axis ($d$), the change in displacement in the $z$ axis direction ($\Delta h$) and the change in helix phase ($\Delta\theta$; Fig. 1b). The remaining degrees of freedom for the positions of helices $b_0$, $c_0$ …, which define the internal geometry of the repeat, are extensively sampled, sequences are designed using Rosetta FastDesign or ProteinMPNN[19,20], and designs are selected for experimental characterization on the basis of packing and sequence–structure consistency metrics (Methods). We obtained synthetic genes encoding the selected designs, expressed them in *Escherichia coli* and purified the proteins using nickel–nitrilotriacetic acid immobilized metal affinity chromatography. Designs that were solubly expressed were analysed by size-exclusion chromatography (SEC) to determine oligomerization state, and in the case of assemblies a subset was analysed by negative-stain electron microscopy (ns-EM). Experimental success rates and structural homogeneity for different classes of designs are summarized in Supplementary Figs. 1 and 2 and Supplementary Discussion.

To generate straight, linear THRs, we set $\Delta\theta$ to zero. As illustrated in Fig. 2a,b, this results in perfectly straight repeat proteins in which each repeat unit is translated but not rotated relative to the previous unit. There are two subclasses: setting $\Delta h = 0$ generates repeat proteins with each repeat unit simply displaced in the $x$–$y$ plane (Fig. 2a), whereas setting $\Delta h$ to a non-zero value generates repeat proteins that also step along the $z$ axis (Fig. 2b). We tested 33 linear THRs (with $\Delta h = 0$) with helices either about 20 or about 40 residues in height; 23 of 33 tested designs were solubly expressed, and 13 of 19 designs analysed by SEC were primarily monomeric as designed (Supplementary Figs. 1a,b and 2). Structural characterization of the linear building blocks by X-ray crystallography individually and/or cryogenic EM (cryo-EM) in the context of assemblies (see below) revealed that both the detailed internal structures and the overall straight linear geometry were successfully achieved. The backbone root mean square deviations (RMSDs) between the design models and crystal structures of three 20-residue helix designs (THR1, THR2 and THR3) and two 40-residue helix designs (THR5 and THR6) were 0.8, 0.8, 0.4, 0.6 and 1.3 Å, respectively, and in all five cases the relative rotation of successive repeats is nearly zero (Fig. 2a and Supplementary Fig. 6a). We found that we could not only control $\Delta\theta = 0$, but also program values of the inter-repeat distance $d$: the crystal structure of a design with $d$ set to a compact helix packing value of 8.7 Å had a very close value of 8.6–8.8 Å at its central interior (THR3), in contrast to most others designed at 10.0 Å (Supplementary Fig. 6b). For structural validation of blocks with non-zero $\Delta h$, the cryo-EM structure of an assembly constructed from such a block (THR4) exhibited a linear stair-stepping structure nearly identical to the design model, (backbone RMSD of 1.0 Å; Fig. 2b and Supplementary Fig. 1a).

To generate turn blocks, we blueprint an additional helix $c_0$ lined up with $a_0$ and $a_1$ that can be assigned any specified phase difference, which can be utilized in fusion operations to produce a turn that is equal to $\theta_c - \theta_a$ (Supplementary Fig. 5d,e). As for all of the THR blocks described here, because of the ideality of the block construction,

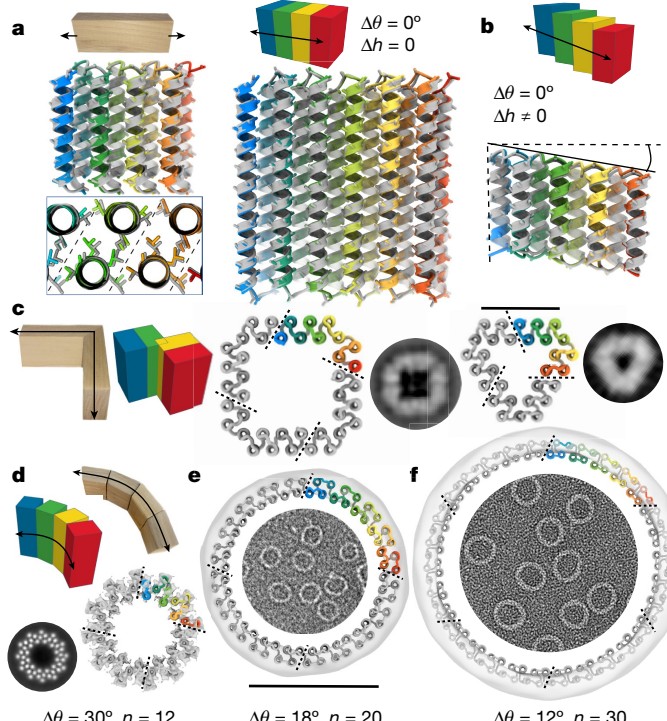

**Fig. 2 | 1D and 2D shapes from THRs. a,b,** The linear THR designs (rainbow) are nearly identical to the experimentally determined structures (grey). Side-chain sticks between α-carbon and β-carbon are shown to indicate helical phasing. **a,** Left: the 2.5-Å-resolution crystal structure of the short, linear THR1 has a 0.8 Å CA RMSD to the design. The inset below shows repeat packing in the THR interior. Right: the 2.7-Å-resolution crystal structure of the tall, linear THR5 has a 0.6 Å CA RMSD to the design. **b,** Bottom: Comparison of the stair-stepping linear THR4 design model to the cryo-EM structure (determined as part of a nanocage assembly; Supplementary Fig. 16). The CA RMSD between the cryo-EM structure and the design model is 1.0 Å. **c,** $C_4$ and $C_3$ polygons generated from four-helix turn module THRs as illustrated on the left. $C_4$ square 90_C4_B (middle) and $C_3$ triangle 120_C3_A (right) oligomers with representative ns-EM 2D class averages for comparison (raw EM micrographs are in Supplementary Fig. 1f). Chain breaks are at the ends of the rainbow sections. Scale bar, 4 nm (for the design models); class averages are not to scale. **d,** Uncapped curve THRs generate cyclic ring oligomers. The 12-repeat ring design (tested as $C_4$) R12B has a cryo-EM 3D reconstruction overlaid on the model; the two are nearly identical. A 2D class average with the individual straight helices resolved is shown left of the ring. **e,** The 20-repeat ring design (tested as $C_4$) R20A has an ns-EM reconstruction density overlaid on the model, and a raw micrograph is shown inside. Scale bar, 10 nm. **f,** The 30-repeat ring design (tested as $C_6$) R30A represented in a similar manner to **e.** Scale bar in **e,** 10 nm (for the design models with reconstruction maps overlaid in **d–f**); class averages are not to scale. The asymmetric unit is coloured in rainbow.

the same sequence interactions can be used for the intra-block and inter-block interactions; we refer to blocks in which the terminal repeats have identical sequences to the internal repeats as uncapped, and those in which the terminal helices have polar outward-facing residues to prevent self-association (like the linear blocks above) as capped. We experimentally characterized uncapped turn modules that generate rotations of 360/$n$, in which $n$ is 3, 4, 5 or 6; if the geometry is correct, these should oligomerize to form closed polygons with $n$ subunits. ns-EM 2D class averages of the $n = 3$ designs clearly show the designed triangular shape with flattened corners (Fig. 2c and Supplementary Fig. 1f), and for $n = 4$, the designed square shapes (Fig. 2c and Supplementary Fig. 1f) including fine details such as the lower density around the corner helix are observed. For $n = 5$ and $n = 6$, success rates were lower, probably because their hinge regions involved less-extensive

helix–helix interactions, but we did obtain designs with the expected polygonal structures for both after using reinforced corners on the $C_6$ design (Supplementary Fig. 1f and Supplementary Discussion). Thus, by controlling the phase rotations between adjacent helices, turns can be encoded while maintaining overall parallel helical architecture. We also made polygonal designs with combinations of linear THRs and new straight helix-heterodimer corner junctions instead of turn modules (Supplementary Discussion and Supplementary Figs. 1g, 9 and 10).

To generate curve THRs, we incorporate a phase change (Δθ) between repeating elements (Fig. 1c) that generates a curved trajectory rather than a linear one. We choose Δθ to be a factor of 360° so that perfectly closed rings can be generated. The size of the closed ring can be controlled by specifying Δθ and the distance $d$ between repeats (Supplementary Fig. 7). To access a broad range of $d$ parameter values, we add additional helices to the repeat unit; for circular rings we used four helices per repeat unit. A full curve THR ring with $n$ repeats can be divided into smaller chains each with $m$ repeats, in which $m$ is a factor of $n$; $n/m$ uncapped repeats can associate to generate the full ring with cyclic symmetry[21]. To facilitate gene synthesis and protein production, we characterized such split oligomeric versions of the rings rather than synthesizing very long single chains. We designed rings with 12, 18, 20 and 30 repeats ranging from 9 to 22 nm in outside diameter. The 12- and 20-repeat rings were tested as $C_4$ designs, whereas the 18- and 30-repeat rings were tested as $C_6$ designs. Designs for all four ring sizes were remarkably uniform with ns-EM micrographs densely covered with circular assemblies with few to no defects or alternative structures present (Supplementary Fig. 7). Two-dimensional class averages showed that designs for all four sizes were close to the intended size (Fig. 2d; 10, 1 and 9 unique designs yielded distinct ring shapes for 18-, 20- and 30-repeat rings (Supplementary Figs. 1e and 2)). The smallest rings with 12 repeats have solvent-exposed helices exterior to the ring placed to facilitate outward-facing fusions without disrupting the core packing of the ring; these are clearly visible in the 2D class averages and 5.2-Å-resolution cryo-EM reconstruction of R12B (Fig. 2d and Supplementary Fig. 1e) that matches the designed patterning of the helices. ns-EM of the 18-, 20- and 30-repeat rings (with outside diameters of 12, 14 and 22 nm respectively) showed that many designs formed remarkably monodisperse populations of ring-like structures closely consistent with the design models (Fig. 2e,f and Supplementary Fig. 1e). ns-EM class averages of these designs had the smooth and round shape of the design models, and were in most but not all cases homogeneous (some designs assembled into closed-ring species that ranged by ±1 chain of the desired number, resulting in some slightly oblong shapes; Supplementary Fig. 1e). These designs highlight the control over ring curvature that can be achieved by specifying building block repeat rotation parameters.

The simplicity of our blocks in principle enables the reinforcing of designed materials using struts rigidly linking distinct structural elements. As a first test of this, we sought to build concentric ring assemblies from pairs of rings that have different sizes but repeat numbers that share large common denominators. For example, 2 repeat units of a 20-repeat ring can be combined with 3 repeat units of a 30-repeat ring as 10 copies generate a complete ring in both cases (Fig. 3a, left). Rings were segmented into matching cyclic symmetries, the rotation and $z$ displacement of one ring relative to the other was sampled, and linear THRs were placed to connect the inner and outer rings. We constructed single-component $C_{10}$ concentric ring assemblies by connecting a three-repeat-unit curved block and a two-repeat-unit curved block that both generate a 36° (360°/10) rotation with a radially oriented strut. Two-dimensional class averages of ns-EM images of the designed strutted assemblies show both rings clearly present (Fig. 3a, right; some 11-subunit rings were observed in addition to the target 10-subunit structure). We similarly connected three repeat units with a 20° rotation per repeat, and five repeat units with a 12° rotation per repeat, with a radial strut; the resulting composite subunits map out

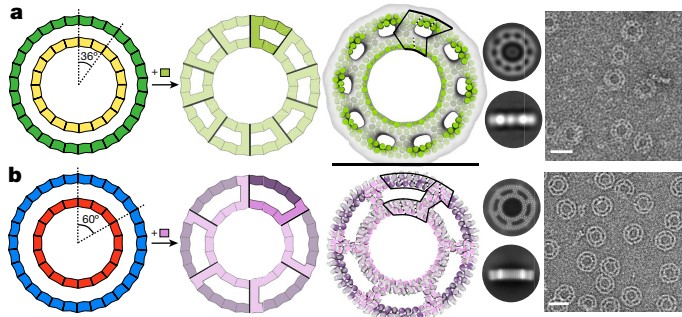

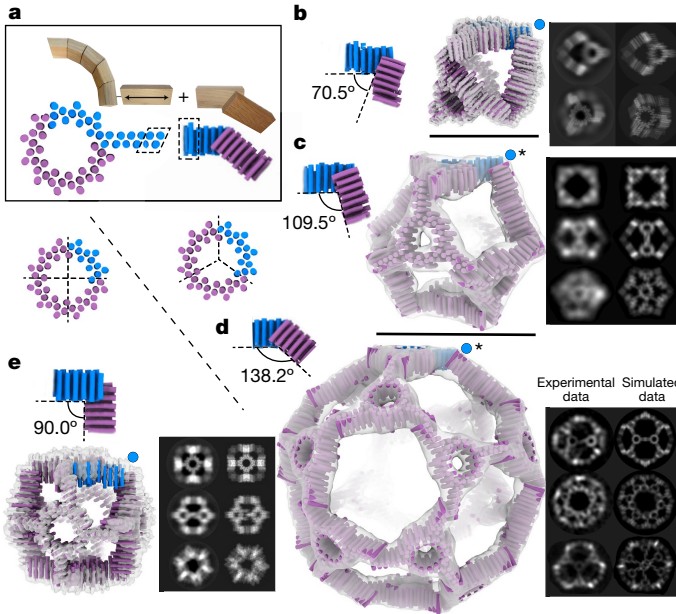

**Fig. 3 | Design of strutted double rings. a,b,** Two different size rings built from curve THRs for which integral multiples generate the same rotation can be concentrically nested and connected by struts. **a,** Three repeats of an outer ring (12° per repeat) are combined with two repeats of an inner ring (18° per repeat) that both generate a 36° rotation. Connection of the two pieces with a linear THR generates a $C_{10}$ single-component ring (strut_C10_8); an asymmetric unit is highlighted in the second ring image. An ns-EM 3D reconstruction in $C_{10}$ symmetry is shown overlaid with the design model next to 2D class averages and a representative micrograph. **b,** Five repeats of an outer ring (12° per repeat) are combined with three repeats of an inner ring (20° per repeat) that both generate a 60° rotation. Connection of the two pieces with a linear THR and an additional chain break in the outer ring generates a $C_6$ two-component ring (strut_C6_21); an asymmetric unit is highlighted in the second ring image with the two chains in different colors. A cryo-EM 3D reconstruction in $C_6$ symmetry is shown overlaid with the design model next to cryo-EM 2D class averages and a representative ns-EM micrograph (additional cryo-EM details are provided in Supplementary Fig. 8c). Scale bars, 20 nm (**a,b**). An asymmetric unit is outlined on top of the design model, and repeats are sectioned with dashed lines.

a 60° rotation of inner and outer rings such that six subunits generate a full 360° ring. The resulting two-component $C_6$ strutted assembly yielded 2D class averages that showed both rings with all chains present, and a 5.1-Å cryo-EM reconstruction was very close to the design model (RMSD 2.7 Å) with very similar outer diameter (19.7 nm versus 20.1 nm; Fig. 3b and Supplementary Fig. 8c). The helix positioning in the inner ring and the strut are also very close to the design model (Supplementary Fig. 8c, insets). Thus, the modularity of the THRs enables designing complex structures by inspection, and enables buttressing to increase structural robustness (Supplementary Discussion and Supplementary Fig. 8).

## Expandable nanomaterials

The regularity of our blocks in principle enables scaling the size of nanomaterial designs simply by changing the number of repeats in the constituent THRs without altering any of the inter-block interfaces. How the THRs must be aligned to enable expandability differs for each architecture, as described below.

To construct expandable cyclic assemblies, the linear THRs must be placed such that the propagation axis is normal to the cyclic symmetry axis. For cyclic designs with this property (those built from turn modules or heterodimers; see Supplementary Discussion), adding or removing repeats simply changes the length of the oligomer edge without affecting the interface between monomers. We tested this expandability with a $C_4$ 'square' (sC4) oligomer for which we had obtained a cryo-EM reconstruction with 1.6-Å-backbone RMSD (Supplementary Fig. 10). This subunit consists of a central linear THR flanked by straight-helix heterodimers that produce a 90° turn. To expand this structure, we inserted two additional repeat units (six helices) into the linear THR portion of the subunit. Cryo-EM 2D class averages for both the original and expanded square show close agreement to the design models and clear expansion; the helices clearly remain aligned to the z axis as designed (Fig. 5a).

**Fig. 4 | Modular construction of protein nanocages from THRs. a,** In box: regular nanocages are constructed from curve THR rings with linear arms projecting outwards that can be linearly extended (left) and designed handshake $C_2$ interfaces (right) that hold the rings at the angle required for the desired polyhedral symmetry. THR chains that are fused into one chain are shown in blue, with identical backbone and sequence areas used for concatenation indicated in dotted outlines. **b–e,** $C_2$ handshake modules generating the required angles (top left in each panel, **b–e**) are combined with either $C_3$ (**b–d**) or $C_4$ (**e**) versions of the ring (shown below the box in **a**) to construct nanocages in **b–e**. Design models are shown as helix cylinders, with the asymmetric unit in blue, and the remaining copies in purple. Each cage is overlaid with the 3D cryo-EM reconstruction or ns-EM (marked with an asterisk here and in subsequent figures) reconstruction, with representative paired 2D classes on the right (left is experimental; right is computed from design model). The blue dots indicate the location of the handshake angle in the cage. **b,** The $T_3$ tetrahedral design cage_T3_101 uses a $C_3$ ring and a 70.5° $C_2$ handshake. **c,** The $O_3$ octahedral design cage_O3_20 uses a $C_3$ ring and a 109.5° $C_2$ handshake. **d,** The $I_3$ icosahedral design cage_I3_8 uses a $C_3$ ring and a 138.2° $C_2$ handshake. **e,** Th $O_4$ octahedral design cage_O4_34 uses a $C_4$ ring and a 90.0° $C_2$ handshake. Scale bars, 20 nm (**b**), 27 nm (**c**), 52 nm (**d**) and 22 nm (**e**).

Architectures with polyhedral nanocage symmetry can be similarly expanded provided that the linear THR propagation axis is parallel to the plane formed by the two symmetry axes spanned by the THR (Supplementary Fig. 11a). To generate such architectures, and enable further access to construction in three dimensions, we designed out-of-plane interactions between building blocks. We first focused on designing $C_2$ symmetric interfaces in which the angles between linear THRs correspond to the angles needed to generate regular polyhedral symmetry (Fig. 4) when combined with planar $C_3$ or $C_4$ components, while also satisfying expandability criteria. For an octahedral 'cube' ($O_4$) built from flat objects with $C_4$ symmetry that lie on the 'cube faces', this angle is 90°. For tetrahedra ($T_3$), octahedra ($O_3$) and icosahedra ($I_3$) built from flat $C_3$-symmetric objects, the out-of-plane handshake angles that are needed to join the flat objects are 70.5°, 109.5° and 138.2°, respectively[22,23]. Handshake $C_2$ homodimers were generated by fixing this out-of-plane angle and keeping the linear THR propagation axes parallel to each other, sampling only the offset spacing between the THRs[8] (Supplementary Fig. 12).

To generate expandable nanocages, flat cyclic components that form the faces of the cages were linked through noncovalent handshake interactions at the specified angle. For the flat cyclic component, we used a ring design with 12 repeats (R12B; Fig. 2c) constructed

from curve units, and split the 12 repeats into either 3 subunits with 4 repeats each ($C_3$) or 4 subunits with 3 repeats each ($C_4$; Supplementary Fig. 3d), depending on the desired polyhedral symmetry architecture. We then fused linear THR arms onto each subunit constrained to point outward parallel to a radial vector emanating from the symmetry axis, but offset such that when the $C_2$ interface is formed, the $C_2$ axis is along a radial vector (Fig. 4a and Supplementary Fig. 12). Tetrahedral, octahedral 'cubic' and icosahedral structures with $C_3$ rings at respective axes ($T_3$, $O_3$ and $I_3$), and octahedral structures with $C_4$ rings at the respective axes ($O_4$) were constructed by incorporation of the appropriate $C_2$ interface. For example, to make a 'cubic' octahedral nanocage, we incorporate into the $C_4$ ring arm the 90° $C_2$ handshake module (Fig. 4e) by simple sequence concatenation. Synthetic genes were obtained for 13 nanocage designs; all 13 expressed solubly, 10 had SEC elution profiles that suggested cage formation, 8 yielded particles with the expected size by ns-EM, and 7 gave 2D class averages and symmetric 3D reconstructions that resembled the design models. Successful designs for each architecture are shown in Fig. 4b–e and Supplementary Figs. 1j and 12. Designed geometric features including the spindle-like two-fold handshake interface and the flat 'in-plane' ring areas with distinct holes are clearly evident. For the $T_3$ and $O_4$ cages, the correct species dominated, but in $O_3$ and $I_3$ cages there were noticeable populations of species that were either partially formed or broken under ns-EM conditions (Supplementary Fig. 13). A 7.5-Å cryo-EM reconstruction and an experimental model were obtained for the cubic cage built from tetrameric rings on the faces (cage_O4_34) that were very close to the design model, with the straight helices clearly evident and only very slight deviations in the arm alignment (Fig. 4e and Supplementary Fig. 14). A 4.0-Å cryo-EM reconstruction and an experimental model for the tetrahedral cage_T3_101 were similarly very close (Fig. 4b and Supplementary Fig. 41). These results illustrate the robustness of structures that can be assembled from our regularized building blocks using simple 'snapping together' of complementary pieces in three dimensions, and show that with additional reinforcing mechanisms such as cooperativity, structural specificity can be achieved without traditional 'knob-and-hole' helix–helix interactions[24].

We tested the expansion in all three dimensions of the cubic design (Fig. 4e and Supplementary Fig. 14) by increasing the number of repeat units in the linear arm. We generated four different sizes of the cage_O4_34 by increasing the number of THR helices in the arm by +0, +4, +8 or +12 helices (Fig. 5b and Supplementary Fig. 13). For all sizes, ns-EM 2D class averages (Fig. 5b, bottom row) show all three symmetrical views with the designed increases in size but otherwise close preservation of architecture. Three-dimensional ns-EM reconstructions were consistent with corresponding design models, with the overall cube shape and ring circular pore clearly visible in each of the sizes (Fig. 5b, top row). The first three sizes of cage show primarily intact assemblies across the ns-EM grids; for the largest size (+12), some incomplete assembly was also observed (Supplementary Fig. 13). Additional single-component expandable nanocage designs are described in Supplementary Figs. 13, 16 and 17 and the Supplementary Discussion.

We next designed two-component expandable nanocages by locking the rotation degrees of freedom of a THR-containing building block to maintain the expandability constraint (Methods), and then docking it against a freely sampling partner oligomer to form an $O_{43}$ architecture (Fig. 5c, Supplementary Discussion and Supplementary Figs. 19–23). Expandability over four different sizes was achieved with cage_O43_129 (+0, +4, +8 and +12 helices). The internal structure of the oligomers is clearly resolved in cryo-EM reconstructions for the first three sizes and in ns-EM reconstruction of the largest size; the distance between the centre of mass of the tetramer component to the centre of mass of the trimer component across the different sizes is 7.9, 9.4, 11.3 and 11.7 nm respectively (Fig. 5c and Supplementary Fig. 21). Views down each of the three symmetry axes (twofold, threefold and fourfold) are clear for each size (except for the threefold view in the largest size) with slight

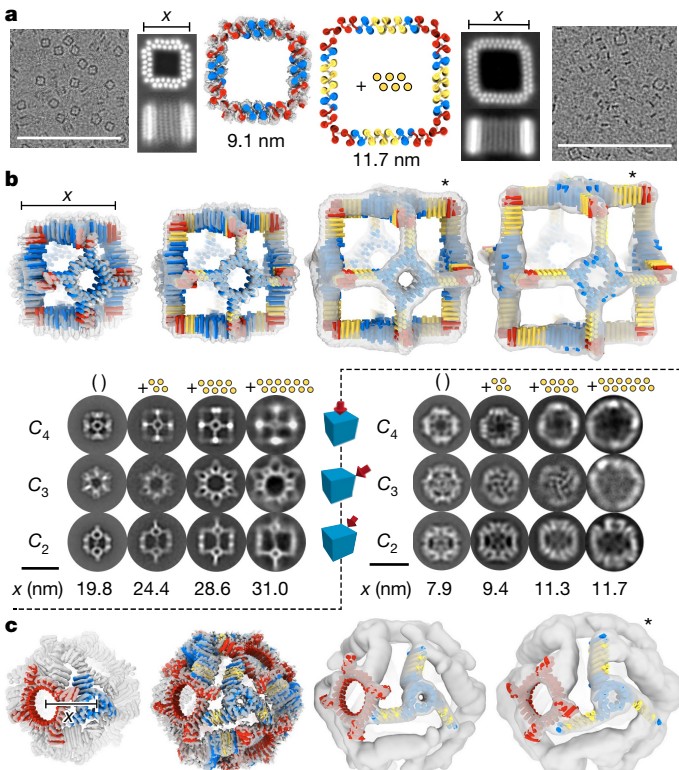

**Fig. 5 | Extendable THR-based nanomaterials. a**, An extendable $C_4$ square. Left: the base design (sC4); right, an expanded version with six additional helices per chain (sC4_+6). Far left and right: representative cryo-EM micrographs (scale bars, 100 nm) with adjacent 2D class averages. A 3D reconstruction is superimposed on the base size design model. Chain breaks are in the red heterodimer region. The length of the side of the square (bar labeled $x$) is indicated beneath the 2D class averages. **b**, Expandable $O_4$ octahedral handshake nanocage, cage_O4_34. Top left to right: design models and cryo-EM or ns-EM reconstructions (the asterisk denotes ns-EM on +8 and +12) of designs extended by 0, 4, 8 and 12 helices. Bottom (left): ns-EM 2D class averages along three symmetry axes (rows) for increasing size designs (columns). The inserted helices are shown in yellow for each case. Scale bars, 30 nm (for 2D class averages). The length of a side of each cube ($x$, measured between the outside corners of the handshake) determined from the EM map volumes is indicated beneath the class averages. **c**, An expandable two-component $O_{43}$ nanocage cage_O43_129. Bottom row: the second size (+4; lowest deviation cryo-EM reconstruction) is shown with a symmetrized design model. For the other sizes, individual oligomers were fitted into the EM density (cryo-EM for +0 and +8 sizes; ns-EM for +12 size) so that rotational deviations from the ideal designs can be analysed (Supplementary Fig. 22). The extendable trimer component is in blue and the constant tetramer component is in red. Right, above 3D cage models: ns-EM 2D class averages along the three symmetry axes for each of the four sizes. The dimension $x$ is the experimentally determined distance between the centre of mass of neighbouring $C_4$ and $C_3$ components. Scale bar, 25 nm (for 2D class averages).

rotational deviations of the fourfold cyclic component compared to the design model, whereas the rotation of the threefold cyclic component holding the THR remains unperturbed as designed (Fig. 5c and Supplementary Fig. 22). A fifth size (16 additional helices) assembled into cage-like structures but the populations were too heterogeneous for detailed characterization (Supplementary Fig. 21).

For unbounded architectures that extend along one or more axes, extensibility requires that the linear THR propagation axes be parallel to the extension axes. We constructed an antiparallel assembly with an overall train track shape from THR modules (Fig. 6a). The 'rails' of the track are linear THRs that are uncapped to allow for unbounded linear assembly end-to-end, and $C_2$ 'ties' dock onto branch interfaces

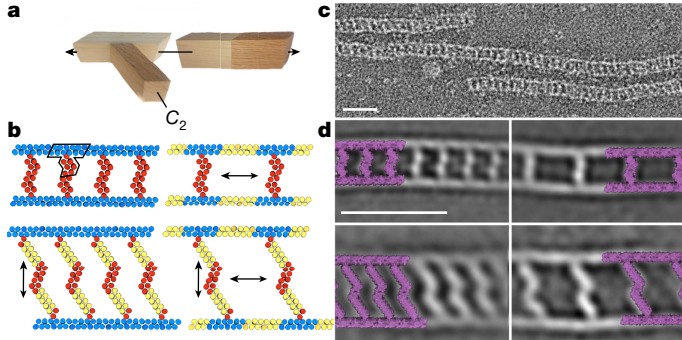

**Fig. 6 | Designed train track fibres. a**, Components of the train track designs: a branch module, a split module and a $C_2$ interface module. **b**, Train track designs. The asymmetric unit is outlined in the top left design; red and blue are unique protein chains. Taking advantage of the extensibility of the building blocks, four different train track designs were created, starting with the design at the top left. Top right: increasing the spacing between rungs by expanding the split module. Bottom left, increasing the length of the rungs by expanding the $C_2$ module. Bottom right: increasing both rung spacing and length by expanding both the split and $C_2$ modules. Inserted segments are in yellow. **c**, An ns-EM micrograph of the design at the top left of **b**. Scale bar, 25 nm. **d**, ns-EM 2D class averages of the four designs in **b**, shown in the same overall layout. The design models are overlaid on the class averages at 1:1 scale. Scale bar, 25 nm.

on the sides of the rails, organizing them into strutted antiparallel pairs. Adding repeats to the rails increases the spacing between ties (along the helical axis) and adding repeats to the ties increases the separation distance between rails along a different axis (Fig. 6b). We used 12-helix addition to the rail to double the spacing between ties, and 8-helix addition to the tie to roughly double the length of the tie. For the four combinations of component sizes, we obtained ns-EM 2D class averages consistent with the design models (compare Fig. 6b and Fig. 6d). Train track assembly was robust to fusion of mScarlet-i on rails both at termini and in an internal loop (Supplementary Fig. 24b), and sfGFP on the ties[25,26], as monitored by ns-EM, with density observed for the GFP, Supplementary Fig. 24c).

## Discussion

On determining the first low-resolution model of the structure of a globular protein (myoglobin), John Kendrew wrote in 1958 that "Perhaps the most remarkable features of the molecule are its complexity and its lack of symmetry. The arrangement seems to be almost totally lacking in the kind of regularities which one instinctively anticipates"[27]. More than six decades of structural biology research have shown this to be a generally appropriate description of protein structure[1]. Figures 2–5 show that this complexity is not an inherent feature of the polypeptide chain: the simplicity and regularity of our designed materials approaches that of the wooden beams used for constructing a house frame. This enables the resizing of designed materials in two and three dimensions simply by changing the numbers of repeat units in the THR modules with little or no need for detailed design calculations; previously this has been possible only with coiled coils and repeat proteins with open helical symmetries (propagating along a single axis)[9,15,28]. The flat surfaces and regular geometry have immediate applications to the design of bio-mineralizing systems: THR monomers presenting carboxylate groups in regular arrays nucleate the mineralization of carbonate into calcite[29], and expandable THR systems such as the cubic assemblies in Fig. 3 presenting such arrays could provide a route to hierarchical protein–mineral hybrid materials.

There are exciting paths forwards to further increase the capabilities of our programmable THR platform. First, our current multi-subunit assemblies have high symmetry, and assembly of arbitrary nanostructures would require breaking symmetry—one approach to achieving this would be to build heterodimeric and heterotrimeric interfaces between THRs, which would enable considerable shape diversification and addressability of each protein chain[30]. This would allow access to a broad range of asymmetric nanostructures, as with DNA nanotechnology bricks, tiles and slats[31–35], but with the higher precision and greater functionality of proteins. Second, the materials generated here all form through self-assembly, but as the number of components increases the overall yield of the desired product could decrease. This limitation could potentially be overcome by stepwise solid-phase assembly with crosslinking after addition of each THR component (as in solid-phase peptide or DNA synthesis, but in three dimensions with the location of addition specified by non-covalent interactions between the THRs; the analogue in construction is nailing lumber pieces together after alignment). The combination of symmetry breaking and stepwise assembly would enable the design of a very wide range of protein nanomaterials based on simple geometric sketches that could be readily genetically modified to present a wide variety of functional domains in precisely controllable relative orientations.

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

# Methods

Computational and experimental methods are all provided in the Supplementary Information.

### Reporting summary

Further information on research design is available in the Nature Portfolio Reporting Summary linked to this article.

## Data availability

All data and design models are available in the main text or the Supplementary Information. EM maps have been deposited in the Electron Microscopy Data Bank (R12B: EMD-43318; strut_C6_21: EMD-29893; cage_O4_34: EMD-29915; cage_O4_34_+4: EMD-41907; cage_O43_129: EMD-42906; cage_O43_129_+4: EMD-42944; cage_O43_129_+8: EMD-42031; sC4: EMD-29974; cage_T3_101: EMD-41364; cage_O3_10: EMD-40070 ($C_1$ asymmetric) and EMD-40071 (octahedral symmetric); cage_T3_5: EMD-40075 ($C_1$ asymmetric), EMD-40074 (tetrahedral symmetric), EMD-40073 ($C_1$, 1 chain missing) and EMD-40073 (1 trimer missing); cage_T3_5_+2: EMD-40076). Crystallographic datasets and cryo-EM structures with resolved side chains have been deposited in the Protein Data Bank (THR1: 8G9J; THR2: 8G9K; THR5: 8GA7; THR6: 8GA6; sC4: 8GEL; cage_T3_101: 8TL7; cage_O43_129: 8V2D; cage_O43_129_+4: 8V3B).

## Code availability

An example RosettaScripts script and input for generating THR building blocks are provided at https://github.com/tfhuddy/2023-manuscript-materials. The example script was confirmed to successfully run with Rosetta version 3.13 as available at https://rosetta-commons.org/ (ref. 19). Documentation for ProteinMPNN sequence design is available at https://github.com/dauparas/ProteinMPNN (ref. 20). Designs were filtered with AlphaFold 2 available at https://github.com/google-deepmind/alphafold (Supplementary Methods).

**Acknowledgements** We thank F. Busch and V. Wysocki for providing native mass spectrometry experiments that helped debug some of our early designs; J. Decarreau for support in trying out optical microscopy with fibres; J. Quispe, S. Dickinson and V. S. Bhatt for assistance with cryo-EM data collection; L. Milles and B. Wicky for wet lab assistance; D. Hicks, H. Pyles and W. Sheffler for computational assistance; S. Boyken, C. Hague, J. Bai and L. Stewart for perspective and discussion; F. Praetorius for manuscript editing; the Arnold and Mabel Beckman Cryo-EM Center at the University of Washington for electron microscope use; and the Advanced Light Source (ALS) beamlines 8.2.1, 5.0.3 and 8.2.2 at Lawrence Berkeley National Laboratory for X-ray crystallography data collection. The Berkeley Center for Structural Biology is supported in part by the National Institutes of Health (NIH), National Institute of General Medical Sciences and the Howard Hughes Medical Institute. The ALS is supported by the Director, Office of Science, Office of Basic Energy Sciences and US Department of Energy (DOE; DE-AC02-05CH11231). This work was supported by the Institute for Protein Design Breakthrough Fund, for "De novo design of 100 nm scale protein assemblies" (T.F.H., Y.H., R.D.K. and D.B.) and "De novo design of selective pores" (Y.L. and D.B.); The Audacious Project at the Institute for Protein Design (T.F.H., J.X., E.C.Y., A.J.B., H.L.H., Z.L., R.M., A.K. and D.B.); The Open Philanthropy Project Improving Protein Design Fund (Y.H., R.R., P.J.Y.L., A.K.B., D.E., G.B. and D.B.); National Science Foundation award CHE-1629214 (D.N. and D.B.); the Helen Hay Whitney Foundation (S.J.C.); a gift from Microsoft (J.D. and D.B.); The Donald and Jo Anne Petersen Endowment for Accelerating Advancements in Alzheimer's Disease Research (T.J.B. and D.B.); and the Howard Hughes Medical Institute (N.B., A.C., B.C. and D.B.). Small-angle X-ray scattering data were collected at the ALS SIBYLS beamline on behalf of US DOE-BER, through the Integrated Diffraction Analysis Technologies programme. Some of the cryo-EM data were collected on a Glacios TEM (Thermo Fisher Scientific) from NIH award S10OD023476 (J.M.K.). Parts of the cryo-EM work were supported by an Open Philanthropy subcontract via the University of Washington. Parts of the cryo-EM data processing were supported by the High Performance Computing facility at NYU School of Medicine. Some of the cryo-EM grids were screened at the Cryo-Electron Microscopy Laboratory Core at NYU School of Medicine (RRID: SCR_019202) and we thank the cryo-EM core staff for their assistance. Some of the cryo-EM data acquisition was carried out at the Simons Electron Microscopy Center and National Resource for Automated Molecular Microscopy and the National Center for cryo-EM Access and Technology located at the New York Structural Biology Center, supported by grants from the Simons Foundation (SF349247) and the NIH National Institute of General Medical Sciences (GM103310, U24 GM129539). This research used resources of the National Energy Research Scientific Computing Center, a US Department of Energy Office of Science User Facility located at Lawrence Berkeley National Laboratory, operated under contract number DE-AC02-05CH11231 using the National Energy Research Scientific Computing Center award BER-ERCAP0022018. This work was supported by the grant DE-SC0018940 MOD03 funded by the US Department of Energy, Office of Science.

**Author contributions** Conceptualization: T.F.H., Y.H., T.J.B., D.B., R.D.K., J.X., D.N., P.J.Y.L. Methodology: T.F.H., Y.H., J.X., R.D.K., P.J.Y.L., E.C.Y., B.C., T.J.B., J.D. Investigation: T.F.H., Y.H., J.X., R.D.K., N.B., D.N., R.R., P.J.Y.L., C.W., S.J.C., A.C., A.J.B., F.A.D.-H., A.K.B., H.L.H., K.D.C., Z.L., R.M., G.R., A.K., B.S., M.S.D., Y.L. Visualization: T.F.H., Y.H., R.D.K., N.C. Funding acquisition: D.B., G.B., D.E., T.F.H., R.D.K., Y.H., A.J.B. Supervision: D.B., G.B., J.M.K., D.E. Writing (original draft): T.F.H., Y.H. Writing (review and editing): T.F.H., D.B., Y.H., R.D.K., E.C.Y., A.K.B., G.B., P.J.Y.L., S.J.C., N.C.

**Competing interests** T.F.H., Y.H., R.D.K. and J.X. are inventors on a provisional patent application submitted by the University of Washington for the design and composition of the proteins created in this study.

**Additional information**
**Correspondence and requests for materials** should be addressed to David Baker.

# Reporting Summary

## Statistics

For all statistical analyses, confirm that the following items are present in the figure legend, table legend, main text, or Methods section.

| n/a | Confirmed | |
|---|---|---|
| ☐ | ☒ | The exact sample size (*n*) for each experimental group/condition, given as a discrete number and unit of measurement |
| ☐ | ☒ | A statement on whether measurements were taken from distinct samples or whether the same sample was measured repeatedly |
| ☒ | ☐ | The statistical test(s) used AND whether they are one- or two-sided<br>*Only common tests should be described solely by name; describe more complex techniques in the Methods section.* |
| ☒ | ☐ | A description of all covariates tested |
| ☒ | ☐ | A description of any assumptions or corrections, such as tests of normality and adjustment for multiple comparisons |
| ☒ | ☐ | A full description of the statistical parameters including central tendency (e.g. means) or other basic estimates (e.g. regression coefficient) AND variation (e.g. standard deviation) or associated estimates of uncertainty (e.g. confidence intervals) |
| ☒ | ☐ | For null hypothesis testing, the test statistic (e.g. *F*, *t*, *r*) with confidence intervals, effect sizes, degrees of freedom and *P* value noted<br>*Give P values as exact values whenever suitable.* |
| ☒ | ☐ | For Bayesian analysis, information on the choice of priors and Markov chain Monte Carlo settings |
| ☒ | ☐ | For hierarchical and complex designs, identification of the appropriate level for tests and full reporting of outcomes |
| ☒ | ☐ | Estimates of effect sizes (e.g. Cohen's *d*, Pearson's *r*), indicating how they were calculated |

*Our web collection on statistics for biologists contains articles on many of the points above.*

## Software and code

Policy information about availability of computer code

| Data collection | Rosetta software (https://rosettacommons.org); WORMS software (https://github.com/willsheffler/worms); RPXDock (https://github.com/willsheffler/rpxdock); ProteinMPNN (https://github.com/dauparas/ProteinMPNN)l Python 3.8; PyRosetta; EPU (FEI Thermo Scientific) |
|---|---|
| Data analysis | CryoSPARC v3; UCSF ChimeraX 1.4; Relion v.3; Coot 0.9.4.1; Phenix 1.13; PyMOL2; MolProbity 4.02b; Namdinator |

For manuscripts utilizing custom algorithms or software that are central to the research but not yet described in published literature, software must be made available to editors and reviewers. We strongly encourage code deposition in a community repository (e.g. GitHub). See the Nature Portfolio guidelines for submitting code & software for further information.

## Data

Policy information about availability of data

All manuscripts must include a data availability statement. This statement should provide the following information, where applicable:
- Accession codes, unique identifiers, or web links for publicly available datasets
- A description of any restrictions on data availability
- For clinical datasets or third party data, please ensure that the statement adheres to our policy

All data are available in the main text or the supplementary materials. EM maps have been deposited in the Electron Microscopy Data Bank (R12B: EMD-43318, strut_C6_21: EMD-29893, cage_O4_34: EMD-29915, cage_O4_34_+4: EMD-41907, sC4: EMD-29974, cage_T3_101: EMD-41364, cage_O3_10: EMD-40070 (C1 asymmetric) and EMD-40071 (octahedral symmetric), cage_T3_5: EMD-40075 (C1 asymmetric) and EMD-40074 (tetrahedral symmetric) and EMD-40073 (C1, 1

chain missing) and EMD-40073 (1 trimer missing), cage_T3_5_+2: EMD-40076). Crystallographic datasets and cryo-EM structures with resolved sidechains have been deposited in the PDB (THR1:8G9J, THR2: 8G9K, THR5:8GA7, THR6: 8GA6, sC4: 8GEL, cage_T3_101: 8TL7). An example script and input for generating THR building blocks are provided at (https://github.com/tfhuddy/2023-manuscript-materials).

REVIEW: Validation reports for review of crystal structures currently erroneously show molecule names with "SRP" prefix instead of "THR"; this will be fixed upon release when new reports are made.

# Research involving human participants, their data, or biological material

Policy information about studies with human participants or human data. See also policy information about sex, gender (identity/presentation), and sexual orientation and race, ethnicity and racism.

| | |
|---|---|
| Reporting on sex and gender | *Use the terms sex (biological attribute) and gender (shaped by social and cultural circumstances) carefully in order to avoid confusing both terms. Indicate if findings apply to only one sex or gender; describe whether sex and gender were considered in study design; whether sex and/or gender was determined based on self-reporting or assigned and methods used.*<br>*Provide in the source data disaggregated sex and gender data, where this information has been collected, and if consent has been obtained for sharing of individual-level data; provide overall numbers in this Reporting Summary. Please state if this information has not been collected.*<br>*Report sex- and gender-based analyses where performed, justify reasons for lack of sex- and gender-based analysis.* |
| Reporting on race, ethnicity, or other socially relevant groupings | *Please specify the socially constructed or socially relevant categorization variable(s) used in your manuscript and explain why they were used. Please note that such variables should not be used as proxies for other socially constructed/relevant variables (for example, race or ethnicity should not be used as a proxy for socioeconomic status).*<br>*Provide clear definitions of the relevant terms used, how they were provided (by the participants/respondents, the researchers, or third parties), and the method(s) used to classify people into the different categories (e.g. self-report, census or administrative data, social media data, etc.)*<br>*Please provide details about how you controlled for confounding variables in your analyses.* |
| Population characteristics | *Describe the covariate-relevant population characteristics of the human research participants (e.g. age, genotypic information, past and current diagnosis and treatment categories). If you filled out the behavioural & social sciences study design questions and have nothing to add here, write "See above."* |
| Recruitment | *Describe how participants were recruited. Outline any potential self-selection bias or other biases that may be present and how these are likely to impact results.* |
| Ethics oversight | *Identify the organization(s) that approved the study protocol.* |

Note that full information on the approval of the study protocol must also be provided in the manuscript.

# Field-specific reporting

Please select the one below that is the best fit for your research. If you are not sure, read the appropriate sections before making your selection.

☒ Life sciences  ☐ Behavioural & social sciences  ☐ Ecological, evolutionary & environmental sciences

For a reference copy of the document with all sections, see nature.com/documents/nr-reporting-summary-flat.pdf

# Life sciences study design

All studies must disclose on these points even when the disclosure is negative.

| | |
|---|---|
| Sample size | Sample size for each design goal was selected based on many factors such as researchers' approximations of how many designs would be needed to obtain multiple characterizable successes for each effort (function of how many new elements are in the particular set) and how many diverse designs passed computational filtering thresholds for each goal. In all cases, sampling could have been done with finer geometric grid sampling as well as more unique sequences generated for each backbone, leaving room for more similar design successes. |
| Data exclusions | There is no data exclusion in this study. |
| Replication | Nanocage and strutted ring designs that received cryo-EM characterization were often expressed and purified separately from the original experiment that led to nsEM characterization. Similarly, for the proteins which yielded crystal structures, the preparation of sample for crystallization was done independently of initial screening. |
| Randomization | Randomization is not relevant to the study. |
| Blinding | Blinding is not relevant to the study. |

# Reporting for specific materials, systems and methods

We require information from authors about some types of materials, experimental systems and methods used in many studies. Here, indicate whether each material, system or method listed is relevant to your study. If you are not sure if a list item applies to your research, read the appropriate section before selecting a response.

## Materials & experimental systems

| n/a | Involved in the study |
|-----|----------------------|
| ☒ ☐ | Antibodies |
| ☒ ☐ | Eukaryotic cell lines |
| ☒ ☐ | Palaeontology and archaeology |
| ☒ ☐ | Animals and other organisms |
| ☒ ☐ | Clinical data |
| ☒ ☐ | Dual use research of concern |
| ☒ ☐ | Plants |

## Methods

| n/a | Involved in the study |
|-----|----------------------|
| ☒ ☐ | ChIP-seq |
| ☒ ☐ | Flow cytometry |
| ☒ ☐ | MRI-based neuroimaging |

