## [Peer Review File · Nature]

Manuscript Title: Blueprinting extendable nanomaterials with standardized protein blocks

Reviewer Comments & Author Rebuttals

Reviewer Reports on the Initial Version:

Referees' comments:

Referee #1:

A manuscript by Huddy et al. from the Baker group describes, in my opinion, an important breakthrough in protein design. In this manuscript, they demonstrate that ML-based protein design is able to construct structures that have been previously reserved for DNA nanostructures, and it seemed that it would be too difficult to ever design polypeptide-based bionanostructures of similar complexity. Needless to say, proteins offer a range of functionalities, and the ability to be genetically encoded is their key advantage for use as materials or molecular machines.

The main breakthrough demonstrated in this contribution is the ability to design links between twist-less helix repeats (THR) to adopt seemingly any angle between the chains, which enabled the construction of diverse designed polypeptide shapes. THRs were demonstrated as versatile building modules that enabled to design dimensions of building modules, resembling the possibility to extend coiled coils.

It is important that the results of this report demonstrate an exciting path ahead, with the potential to engineer different types of building modules besides THRs, such as e.g. by the introduction of a twist and an asymmetric assembly, other symmetries as well as asymmetric assemblies. It also seems likely that those building modules will facilitate the preparation of 2D and 3D protein lattices in the future.

An important issue that the authors should specify is the estimated number of possible orthogonal interaction surfaces between THR modules, as this will be a key to the achievable complexity and introduction of asymmetric shapes.

The authors could consider whether the term expandable in the title is most appropriate since it resembles at first glance expandable materials, e.g. in expansion microscopy, where the material is truly expanded and not prepared in different versions with extended dimensions as in this publication. The terms extend and shorten may be more appropriate.

In Fig. 2f and in the text $n=30$ is specified for the largest ring design, although the cartoon highlights 20 helices per asymmetric unit (C6). This should be clarified.

What is the yield of the generated proteins and assemblies, as this is important for technological relevance? Even a low yield would be satisfactory to demonstrate the proof of the principle; however, it would be important for other scientists to know the yield of the final assemblies.

While electron micrographs suggest for most designs that many homogeneous particles were formed, no yield has been specified. For multichain assemblies yields are expected to be lower, but to what extent – 80%, 10%, or 1%? Were the assemblies used for the EM analysis produced from SEC fractions?

Given the strong periodicities of the assemblies solution, SAXS measurements should be expected to feature a clearly discernible signal based on the regularity of the assemblies in bulk solution. SAXS scattering curves are shown for THR modules but not for the assemblies. Were SAXS experiments not performed for others or the results did not agree as well with the prediction as in some other multicomponent assemblies by the same group e.g. in Lutz et al. (Science, 2013, Fig. S16)?

I haven't found in the supplement AA sequences of the successful designs, which are definitely required for publication. This is needed in order to enable others to reproduce the results and to build upon this science.

In conclusion, this is an excellent well-performed, detailed study, which demonstrates the huge potential of de novo protein design.

Author Rebuttals to Initial Comments:

Response to Referee Comments

Referee #1:

A manuscript by Huddy et al. from the Baker group describes, in my opinion, an important breakthrough in protein design. In this manuscript, they demonstrate that ML-based protein design is able to construct structures that have been previously reserved for DNA nanostructures, and it seemed that it would be too difficult to ever design polypeptide-based bionanostructures of similar complexity. Needless to say, proteins offer a range of functionalities, and the ability to be genetically encoded is their key advantage for use as materials or molecular machines.

The main breakthrough demonstrated in this contribution is the ability to design links between twist-less helix repeats (THR) to adopt seemingly any angle between the chains, which enabled the construction of diverse designed polypeptide shapes. THRs were demonstrated as versatile building modules that enabled to design dimensions of building modules, resembling the possibility to extend coiled coils.

It is important that the results of this report demonstrate an exciting path ahead, with the potential to engineer different types of building modules besides THRs, such as e.g. by the introduction of a twist and an asymmetric assembly, other symmetries as well as asymmetric assemblies. It also seems likely that those building modules will facilitate the preparation of 2D and 3D protein lattices in the future.

We thank the reviewer for their detailed review of our manuscript. We believe that the expandable nanocage results in particular are striking examples of how designs can be successful with strict geometric constraints in 3D. This concept can, in principle, be combined with heteromeric blocks to make asymmetric assemblies that still retain geometric constraints, especially if using blocks that are symmetric in backbone shape but asymmetric in sequence (Kibler *et al*, in review:

<https://www.biorxiv.org/content/10.1101/2023.04.07.535760v1>).

An important issue that the authors should specify is the estimated number of possible orthogonal interaction surfaces between THR modules, as this will be a key to the achievable complexity and introduction of asymmetric shapes.

As an example, the strutted rings in Fig. 3 show up to four interfaces per THR chain. In panel A, the single-component ring has high fidelity of correctly forming, where the 2 outer interfaces on either side of the ring form with their intended partners instead of head-to-head or tail-to-tail, and likewise for the two on the inner ring. It is worth noting that these are mainly hydrophobic interfaces with heavy constraints on interface shape (very flat helical interfaces), separated in space by geometry. It is unlikely that they will remain highly orthogonal when there are too many interfaces without polar surfaces to separate them.

It is possible to add more interfaces to the outward facing perimeter to further increase the size and complexity of the ring. To achieve orthogonality of more interfaces with these similar backbone designs we could utilize hydrogen-bond networks in the interfaces (<https://www.nature.com/articles/s41586-018-0802-y>) or could even use powerful new ML-based protein design methods to introduce local deviations from our THR helices to incorporate beta-strand containing interfaces which are highly orthogonal even when several chains are expressed separately and later combined (<https://www.science.org/doi/10.1126/science.abj7662>).

However, since the behavior of each chain in a protein assembly can vary based on if they are all co-expressed together or if each is purified separately, it may be outside the scope of this paper to speculate the quality of behavior of any future alternative methods. Moreover, complications in assembly beyond what is shown could often be due to general biochemistry challenges with balancing kinetics and thermodynamics and not due primarily to design shape or even interface style.

The authors could consider whether the term expandable in the title is most appropriate since it resembles at first glance expandable materials, e.g. in expansion microscopy, where the material is truly expanded and not prepared in different versions with extended dimensions as in this publication. The terms extend and shorten may be more appropriate.

We agree that the perfect wording has been difficult to find. To your point, we think the word “scalable” is more appropriate but that has connotations related to scaling up production yield of materials rather than molecule sizes. For now we will follow your suggestion and substitute in “extendable”.

In Fig. 2f and in the text $n=30$ is specified for the largest ring design, although the cartoon highlights 20 helices per asymmetric unit (C6). This should be clarified.

We have added clarification in the figure legend to address this concern. The figure legend now also lists what symmetry the design was tested as (eg. C6) in addition to the internal repeat count in design (eg. $n=30$).

What is the yield of the generated proteins and assemblies, as this is important for technological relevance? Even a low yield would be satisfactory to demonstrate the proof of the principle; however, it would be important for other scientists to know the yield of the final assemblies.

While electron micrographs suggest for most designs that many homogeneous particles were formed, no yield has been specified. For multichain assemblies yields are expected to be lower, but to what extent – 80%, 10%, or 1%? Were the assemblies used for the EM analysis produced from SEC fractions?

To address these valid concerns, we have provided an additional section in the supplemental discussion to discuss assembly yield, and an additional supplemental figure to support it with experimental Size Exclusion Chromatography (SEC) data which indicates how much of the protein input (IMAC purified as described in methods) ended up in the protein output (selected SEC fractions) for EM analysis.

These sections are appended to the end of this document.

Given the strong periodicities of the assemblies solution, SAXS measurements should be expected to feature a clearly discernible signal based on the regularity of the assemblies in bulk solution. SAXS scattering curves are shown for THR modules but not for the assemblies. Were SAXS experiments not performed for others or the

results did not agree as well with the prediction as in some other multicomponent assemblies by the same group e.g. in Lutz et al. (Science, 2013, Fig. S16)?

SAXS experiments were not performed on THR assemblies. We only attempted SAXS experiments on the tall linear THR modules in Supp Figure S1B, which resulted in 2 designs being dropped from the block set due to poor fit of data, and the rest included as shown. Despite SAXS being able to show key features of many of these designs as the reviewer describes, nsEM was eventually chosen as the preferred validation method for larger constructs because of its ability to reveal multiple shapes if present, and the ability to easily analyze those species.

I haven't found in the supplement AA sequences of the successful designs, which are definitely required for publication. This is needed in order to enable others to reproduce the results and to build upon this science.

These will all be provided upon publication in a supplementary file, as will all of the design models and experimental models that are included in the manuscript.

In conclusion, this is an excellent well-performed, detailed study, which demonstrates the huge potential of de novo protein design.

(End of Reviewer 1 Comments)

Additional paragraph added to supplemental discussion:

Comments on assembly yield

Supplemental Figure S32 is included to give examples of typical assembly yield for protein assemblies from THRs. Successful ring designs, both one-component and two-component, typically oligomerize to near completion, with a single major peak in the SEC traces (fig S32A). Nanocage designs are more varied at larger sizes; SEC traces for the four sizes of *cage_O4_34* are provided to illustrate this (fig S32B). The smaller two sizes show predominantly single species, whereas the larger two sizes have tails to their major peak and additional peaks at higher elution volumes, suggesting partially assembled species. Furthermore, a decrease in the total soluble protein yield is observed as monomer size increases (fig S32B).

Additional supplemental figure added:

Fig. S32. Size Exclusion Chromatography (SEC) traces of representative designs

(A) Overlaid SEC traces of selected ring designs ran on Superdex® 200 Increase 10/300 GL columns after IMAC purification. Absorbance at 230 nm wavelength is normalized between samples because not all were prepared at the same expression scale. *R12A*, *R20A*, and *R30A* are single component multimeric designs, and *strut_C6_21* is a 2-component multimeric design. (B) Overlaid SEC traces for 4 sizes of *cage_O4_34*, non-normalized absorbance at 230 nm wavelength, ran on Superose® 6 Increase 10/300 GL columns. All samples in (B) were prepared from 50 mL culture with autoinduction for expression. Stars indicate the manually determined major peak per SEC trace.

Reviewer Reports on the First Revision:

Referees' comments:

Referee #1:

Authors have adequately responded to all questions, and I find the manuscript suitable for publication.

Referee #4:

In their manuscript Huddy et al. describe the design and experimental validation of linear, curved, and angled protein building blocks as well as inter-block interactions. They use their modular building blocks to design nanomaterials ranging from circular oligomers over polyhedral nanocages to unbounded nanostructures, essentially covering a large geometric space and thoroughly establishing the usefulness of their approach.

This is an extremely large body of work and in my opinion a giant leap forward in the design of protein nanomaterials - bordering design aspects that were previously only achievable via DNA nanostructures.

The authors elegantly show how a combination of parametrically designed helices and clever geometric arrangements yield highly idealized and regular de novo protein building blocks. However, I think Huddy et al. should address a couple of points before the manuscript can be published.

1. Huddy et al. should briefly discuss their ways of sequence design. I believe that they were satisfied with the sequences from ProteinMPNN; however, I find it surprising that fast design was used in the first place for these regular building blocks. Why was a simple pack rotamers never used if backbone ideality was of importance? In addition it would be great to compare design success rates to the chosen sequence design method.

2. The manuscript main text would in my opinion greatly benefit from a summary table that details how many unique straight/curved/angled building blocks and thus recombinable interfaces the authors generated in this work. Even though it might be too much to speculate on how many unique interfaces one could come up with, this will provide the readers with a better grasp of what is possible.

3. For the hand-shake module in particular, I am missing a discussion about possible angles this can have.

4. Table S1b: The q-values are missing from the SAXS curves and the FOXS fits seem very low. It would be great to comment on that and add in the missing x-axis.

5. The R factor values for reported structure THR6 seem rather high for the given resolution, and the

number of water molecules slightly excessive. However the remaining values look good. Can the authors explain why that is? Is part of the this structure disordered?

Referee #5:

The authors introduce a general protein-based design approach to enable the fabrication of a broad range of 2D and 3D protein-based assemblies that do not need to conform to regular, symmetric geometries, nor be constrained to a certain size/scale due to underlying protein building blocks. They validate and explore their framework extensively experimentally using crystallography and single-particle cryo-electron microscopy using a range of geometries spanning their elemental helix-bundle building blocks, closed rings and geometric cavities with programmed curvatures or angles, concentric rings, and polygonal nanocages that are extendable based on underlying modules of arrayed helices.

Overall, the framework goes well beyond existing protein-based fabrication capabilities to approach the capabilities offered by DNA origami for arbitrary geometric 2D and 3D fabrication, with the additional capability of extendable size, demonstrated up to a point. These materials could form the basis of protein-mineral and other hybrid materials that could offer interesting and impactful applications in materials science and engineering.

My only suggestions for minor revision would be first, to ensure that readers are aware that the thermal stability and mechanical properties of these materials may be quite distinct from protein materials fabricated using packed globular monomers (e.g., nanocages designed using Neil King et al.'s approach), given the elemental coiled-coil stacking interactions that may be susceptible to iso-energetic shear and bending deformations. And second, to clarify in Discussion to what extent the design framework will be accessible to other researchers, in particular in contrast to DNA origami algorithms (not referenced) that have disseminated powerful top-down computational algorithms to enable anyone to fabricate such arbitrary 2D and 3D geometries merely from CAD geometries, without any knowledge or training in DNA nanotechnology per se. In this vein, it would be helpful to comment on whether such a top-down computational design framework is already enabled by the current work, or whether this would require substantial additional work to disseminate most broadly the impact of the design strategy to the materials science communities.

Author Rebuttals to First Revision:

Response to Referee Comments

Referee #4:

In their manuscript Huddy et al. describe the design and experimental validation of linear, curved, and angled protein building blocks as well as inter-block interactions. They use their modular building blocks to design nanomaterials ranging from circular oligomers over polyhedral nanocages to unbounded nanostructures, essentially covering a large geometric space and thoroughly establishing the usefulness of their approach.

This is an extremely large body of work and in my opinion a giant leap forward in the design of protein nanomaterials - bordering design aspects that were previously only achievable via DNA nanostructures.

The authors elegantly show how a combination of parametrically designed helices and clever geometric arrangements yield highly idealized and regular de novo protein building blocks. However, I think Huddy et al. should address a couple of points before the manuscript can be published.

1. Huddy et al. should briefly discuss their ways of sequence design. I believe that they were satisfied with the sequences from ProteinMPNN; however, I find it surprising that fast design was used in the first place for these regular building blocks. Why was a simple pack rotamers never used if backbone ideality was of importance? In addition it would be great to compare design success rates to the chosen sequence design method.

We thank the reviewer for the detailed review of our manuscript. The reviewer points out the different methodologies used for sequence design. Historically, designs were made with the Rosetta packer (guided by Rosetta energy unit

optimization, with or without backbone movement during design) before the advent of ProteinMPNN. As ProteinMPNN has benchmarked better both in success rate and especially in computational efficiency, the newer designs utilized the newer design tool instead. We have addressed this switch in the methods section.

We have noted and added the missing RosettaDesign description in the methods section. While PackRotamersMover by itself will perfectly maintain the ideality of the backbone, we observed that very high alanine content with voids was a consistent issue. We hypothesized that this was due to the full atom nature of Rosetta where the repulsive score (fa_rep) heavily penalizes even slight overpacking. The solution was to utilize FastDesign which allowed local backbone movement to breathe away such slight clashes, but was set to heavily limit this behavior to maintain near ideality. This allowed the software to select identities that pack the cores of the models much better.

We agree that design success rates using the different methods would be insightful to compare. However, within the scope of this particular project, we believe that there were too many modified variables (number of components in assembly, geometry, assembly size and level of reinforcement, etc.) aside from sequence design methodology between design rounds to make the data meaningfully comparable.

2. The manuscript main text would in my opinion greatly benefit from a summary table that details how many unique straight/curved/angled building blocks and thus recombinable interfaces the authors generated in this work. Even though it might be too much to speculate on how many unique interfaces one could come up with, this will provide the readers with a better grasp of what is possible.

We have now included a summary table in the supplemental materials to tabulate the number of building blocks per topology.

Summary of designs included in this work			
Figure Section	Design type	Number of entries	Level of characterization
S1A	Linear THRs	6	crystal structure or cryo-EM structure
S1B	Linear THRs	6	SAXS analysis
S1C	Linear THRs	10	ns-EM and cryo-EM structure validation when used as part of an oligomer
S1D	Linear THRs	33	design models
S1E	Curving THRs tested as rings	19	ns-EM structure validation
S1F	Polygon oligomers from Turn module	5	ns-EM 2D class averages
S1G	Other cyclic oligomers	5	cryo-EM and ns-EM structure validation
S1H	THR arm fusions to published oligomer	3	ns-EM 2D class averages
S1I	C2 Handshake designs	4	cryo-EM and ns-EM as part of nanocages
S1J	Nanocage designs	22	cryo-EM and ns-EM structure validation
S1K	Struttred THR designs	7	cryo-EM and ns-EM structure validation
	Total (with some redundancy)	120	

3. For the hand-shake module in particular, I am missing a discussion about possible angles this can have.

In the context of the manuscript, the angles tested were exclusively for the polyhedral symmetries. While it is most likely possible to generate the hand-shake modules for any given angle, the “goodness” of the interface is reinforced by cooperativity of the final architecture in our case. Systems without considerable cooperativity may sometimes require better, more idealized (eg. knob-in-hole) interactions as previously described by DeGrado and/or Woolfson *et al.* In order to confidently not-overclaim on this matter, we have edited a main text sentence to reflect the geometrical bonuses in our system.

We also added a statement in the supplementary methods to reflect that other angles were not directly investigated.

4. Table S1b: The q -values are missing from the SAXS curves and the FOXS fits seem very low. It would be great to comment on that and add in the missing x -axis.

We thank the reviewer for pointing out the missing q values. We assume the low FoXS fits the reviewer is referring to are the reported χ^2 (chi-squared) values.

Low χ^2 values indicate good agreement between experimental data and the model. Extremely low χ^2 values ($\ll 1$) can be a result of very good agreement between the data and the model (a perfect agreement would have χ^2 of 0), but such values can also indicate overfitting or inaccurate estimates of experimental error.

The FoXS server generates theoretical data from the input design model which it fits to the data using three free parameters, c_1 (scaling of atomic radius), c_2 (contribution of hydration layer), and an intensity scaling factor. To avoid overfitting, we re-analyzed the data and fixed c_1 and c_2 to their default values (1.0 and 0.0, respectively), leaving only the scaling parameter. We also modeled the His-tags with the FloppyTail Rosetta application and used this as an additional theoretical model. Generally, the χ^2 values remained low despite some previous overfitting by the FoxS server, with the models with a tag showing lower χ^2 values which indicate that these are better models to explain the data.

For the χ^2 calculation, the error is taken from the standard deviation of the raw experimental data which can be affected by buffer subtraction [doi: 10.1007/978-1-62703-691-7_18]. For these datasets, the χ^2 values for different buffer subtractions are similar to each other so we report the one with the lowest value.

We included for comparison the FoXS fit for THR5, a design for which we obtained a crystal structure with 0.6Å C-alpha RMSD to the design model. This model achieved a χ^2 value of 0.33 (0.27 with the tag), which is among the lowest of the dataset suggesting this is still reasonable χ^2 territory.

The small residuals in the fit between our theoretical model and the THR5 data give us confidence that the SAXS data are well matched to our design models to the resolution that SAXS affords. Since this is not a replacement for more-definitive structural data such as crystal structures or cryo-EM, these designs remain in our supplemental summary table in their own section, but now with an additional full-page supplemental figure (shown below) of higher-quality SAXS plots made with consistent c1 and c2 parameters which should give readers more accurate representation of each design so they can better consider which ones they want to include in any work.

5. The R factor values for reported structure THR6 seem rather high for the given resolution, and the number of water molecules slightly excessive. However the remaining values look good. Can the authors explain why that is? Is part of the this structure disordered?

We appreciate your detailed review. The diffraction data has some anisotropy and the loop regions have some disorder that impact the R-values. We have re-refined the structure to address this. A few side chains were added and some water molecules were removed. The R-values have improved a bit and new coordinates were updated to PDB deposition as well as crystallographic table.

-

Referee #5:

The authors introduce a general protein-based design approach to enable the fabrication of a broad range of 2D and 3D protein-based assemblies that do not need to conform to regular, symmetric geometries, nor be constrained to a certain size/scale due to underlying protein building blocks. They validate and explore their framework extensively experimentally using crystallography and single-particle cryo-electron microscopy using a range of geometries spanning their elemental helix-bundle building blocks, closed rings and geometric cavities with programmed curvatures or angles, concentric rings, and polygonal nanocages that are extendable based on underlying modules of arrayed helices.

Overall, the framework goes well beyond existing protein-based fabrication capabilities to approach the capabilities offered by DNA origami for arbitrary geometric 2D and 3D fabrication, with the additional capability of extendable size, demonstrated up to a point. These materials could form the basis of protein-mineral

and other hybrid materials that could offer interesting and impactful applications in materials science and engineering.

My only suggestions for minor revision would be first, to ensure that readers are aware that the thermal stability and mechanical properties of these materials may be quite distinct from protein materials fabricated using packed globular monomers (e.g., nanocages designed using Neil King et al.'s approach), given the elemental coiled-coil stacking interactions that may be susceptible to iso-energetic shear and bending deformations.

First, we thank the reviewer for their thoughtful review of our manuscript. These concerns are great points. Indeed the iso-energetic alternate conformations of geometrically aligned helical bundles should be a consideration by researchers, and we agree that this warrants comment in the supplemental discussion section (a section has been added). To comment on this here, the bending/rotation of helices around each other has been observed in this work; this is illustrated when our larger designed rings form multiple species- the supplemental figure 1 shows many more rings besides those in the main text, many of which have this feature. Still, it seems that it is possible for these topologies to achieve high monodispersity in some designs- granted some of these display not-perfectly-circular shapes of the correct oligomeric state in some negative stain EM 2D classes. It is hard to say if these are skewed views or how much the negative stain environment has affected them.

Next, we do not have obvious noted examples of “Z-axis” shearing between helices in this manuscript, but this phenomenon is appreciated by researchers in the helical bundle field as you have noted. Particularly if one is using designs that take advantage of extended repeats in helical geometry (7 residue heptad phasing repeat as seen in coiled coils, or 11 and 18 residue phasing repeats in some straighter bundles; this is a different kind of repeat compared to what is described in most of this manuscript), care should be taken to diversify the interactions along the repeated helical lengths to prevent iso-energetic sliding. Hydrogen-bond-network patterning along helical repeats as used by Chen et al in the past (<https://www.nature.com/articles/s41586-018-0802-y>) should be a reliable method to tackle this, which can offer energetic penalties to alternative conformations that are a result of shearing compared to the designed state.

And second, to clarify in Discussion to what extent the design framework will be accessible to other researchers, in particular in contrast to DNA origami algorithms (not referenced) that have disseminated powerful top-down computational algorithms to enable anyone to fabricate such arbitrary 2D and 3D geometries merely from CAD geometries, without any knowledge or training in DNA nanotechnology per se. In this vein, it would be helpful to comment on whether such a top-down computational design framework is already enabled by the current work, or whether this would require substantial additional work to disseminate most broadly the impact of the design strategy to the materials science communities.

To discuss the comparisons to DNA origami algorithms, we would say that we are still not quite ready for a nanoCAD “SolidWorks” style drafting protocol for proteins (at least at the level of complexity we aspire to). We have thought about this much and, for example, if we were to try to make the full house-frame as shown in main text Fig 1, we think it would require optimization of interface interaction orthogonality and could also likely benefit from some crosslinking of intermediate assemblies. However those goals do seem realizable, and the regularized structure is still a big part of what would make this computationally tractable for us.

Still, in the absence of such a hands-off computer program, there exists a wealth of things that researchers can do with these blocks and the associated published protocols for orienting them and making more of them tailor-made. A few immediately pursuable usages include:

(1) the digitally prescribed nature of the block geometries and their assemblies make for ideal integration into efforts to match lattice parameters of minerals as you alluded to,

(2) there are ongoing efforts to make scaffolds with several components across several nanometers that, when combined with different fusion proteins of mini-binders or scFV antibody fragments, could make signaling hubs that approach emulation of immune synapse biology (the nested ring geometries combined with symmetry breaking such as described in another work with similar protein backbones would be a good start for this (<https://www.biorxiv.org/content/10.1101/2023.04.07.535760v1>)), and

(3) making more varieties of protein nanocages for antigen display that are modularly tunable, following on the success from Icosavax with protein nanocage vaccines (<https://www.astrazeneca.com/media-centre/press-releases/2023/astrazeneca-to-acquire-icosavax-including-potential-first-in-class-rsv-and-hmpv-combination-vaccine-with-positive-phase-ii-data.html>).

For electron microscopy technical reviewer:

Improvements made to electron microscopy data

- The tetrahedral cage formed from Handshake module (Fig 4B) has undergone cryo-EM characterization and model building, improving the quality of the structural data compared to the previous negative stain EM. The overlaid map and 2D classes shown in Fig 4B have been updated accordingly. Details of the built model are in supplementary figure S41.
- An additional size of the O4 expanding octahedral cage (Fig 5B) has undergone cryo-EM characterization and model building, improving the quality of the structural data compared to the previous negative stain EM. The overlaid map in Fig 5B has been updated accordingly.
- Three sizes of the O43 expanding two-component cage (Fig 5C, Supp Fig S22) have undergone cryo-EM characterization, improving the quality of the structural data compared to the previous negative stain EM. The overlaid maps in Fig 5C have been updated accordingly. The supplemental figure S22 also shows an improvement in accuracy to the design models compared to the previous negative stain EM. The maps for the +0 and +4 sizes were of high quality, so two additional supplemental figures S42 and S43 were added to show details of the built models.
- We had some missing large-field views of negative stain EM for some of the polygon oligomers (Fig 2C). Additional supplemental figure S33 has been added to include these so the particle dispersion on the grids can be adequately assessed by readers.
- Supplemental figure S35 was added to show additional comparison of the cryoEM model for ring *R12B* compared to design model; previously

there was only data shown in main text Fig 2D which is not as informative.

- **Supplemental Figures showing the processing pipelines for all the new cryo-EM have been appended to the manuscript accordingly.**

Reviewer Reports on the Second Revision:

Referees' comments:

Referee #4:

With their new additions to the main text and extra sections and further clarifications in the supplementary materials, the authors have adequately addressed all my questions and concerns. Therefore, I recommend the publication of the manuscript without any further additions/changes.

Referee #5:

The authors have addressed my comments, so I'm happy to endorse publication of this impactful work.